



# Wildland fire potential outlooks for Portugal using meteorological indices of fire danger

Sílvia A. Nunes[1], Carlos C. DaCamara[1], Kamil F. Turkman[2], Teresa J. Calado[1], Ricardo M. Trigo[1]

[1]Instituto Dom Luiz (IDL), Faculdade de Ciências, Universidade de Lisboa, 1749-016 Lisboa, Portugal
[2]DEIO-CEAUL, Universidade de Lisboa, 1749-016 Lisboa, Portugal

*Correspondence to*: Sílvia A. Nunes (sanunes@fc.ul.pt)

**Abstract**

Portugal is recurrently affected by large wildfire events that have serious impacts at the socio-economic and environmental levels and dramatic consequences associated with the loss of lives and the destruction of the landscape. Accordingly, seasonal forecasts are required to assist fire managers, thus contributing to alter the historically-based purely reactive response. In this context, we present and discuss a statistical model to estimate the probability that the total burned area during summer will exceed a given threshold. The statistical model uses meteorological information that rates the accumulation of thermal and vegetation stress. Outlooks for the 39-year study period (1980-2018) show that, when the statistical model is applied from May 26 to June 30, out of the six severe years, only one year is not anticipated as potentially severe and, out of the six weak years, only one is not anticipated as potentially weak. The availability of outlooks of wildfire potential with an anticipation of up to one month before the starting of the fire season, such as the one proposed here, may serve to provide clear directions for the fire community when planning prevention and combating fire events.

## 1 Introduction

Portugal is regularly affected by very large and destructive wildfires that represent a serious threaten to human lives and to the territory, and have very strong and adverse impacts at the social, economic, ecologic and environmental levels that include human casualties, the destruction of homes and other structures, damages to forests, agricultural fields, shrublands and livestock, changes in the landscape and emission of greenhouse gases (Costa et al., 2010; Pereira et al., 2011). According to the European Commission Technical Report about forest fires in Europe (San-Miguel-Ayanz et al., 2018), from 1980 to 2017, Portugal accounted for 26% of the total burned area in the five southern member states (Portugal, Spain, Italy, Greece and France). However, this figure raises up to 42% when restricting to the last decade, the extreme year of 2017 deserving a special emphasis with Portugal accounting for 59% of the burned area in the five southern member states, despite representing 6% of the total area of the countries.

As in Mediterranean Europe, fire activity in Portugal involves complex interactions among climate, vegetation and humans (Lavorel et al., 2006; Costa et al., 2010). Persistent warm and dry conditions followed by heat spells in summer provide the



optimal meteorological background to the onset and spreading of large fire events (Pereira et al., 2005; Trigo et al., 2006; DaCamara et al., 2014) that take place in a landscape of cumulated biomass and increased fuel connectivity as a result of agricultural abandonment and forest expansion (Pausas and Fernández-Muñoz, 2012; Fernandes et al., 2014; Viedma et al., 2015; Oliveira et al., 2017). In addition, Portugal has the highest density of ignitions in Southern Europe, with the human

presence and activity being the key drivers of ignitions, most of them associated to land management practices and inadequate use of fire (Catry et al., 2009). Finally, the impacts of climate change cannot be disregarded, since a future warmer climate will steer a larger number of severe wildfire episodes (Flannigan et al., 2013; Sousa et al., 2015). In this context, the catastrophic fires of June 2017 in Portugal (with 65 fatalities) associated to the stronger heatwave ever observed over Iberia in June, provide a stark reminder of the growing likelihood of these events under the current warming climate (Sánchez-Benítez

et al. 2017).

The magnitude of the problems related to fire activity in Portugal has motivated the scientific and technical communities to cooperate with forest managers, firefighters and the fire community with the aim of improving the understanding of the fire regime in Portugal and develop better tools that will help preventing and mitigating the impacts of severe fire events (Collins et al., 2013; Oliveira et al., 2016).

Outlooks of wildfire potential for different regions of the globe are usually based on statistical approaches (e.g. Gudmundsson et al., 2014; Turco et al., 2017) or, alternatively, using fire weather predictions a few months in advance based on dynamical seasonal forecasts by atmospheric circulation models (Anderson et al., 2007; Turco et al., 2018). For instance, the European Forest Fire Information System (EFFIS), one of the components of the Emergency Management Services in the EU Copernicus program, is currently disseminating maps (as experimental products) of long-term seasonal forecasts of temperature and

rainfall anomalies based on the European Centre for Medium-Range Weather Forecasts (ECMWF) Seasonal Forecasting System (System 4). However, as pointed out by Bedia et al. (2018), although there is significant skill in predicting one month ahead above average summer fire weather in some parts of south-eastern Europe, skill is in general quite poor elsewhere.

As an alternative, the aim of this paper is to describe and discuss a model that allows making outlooks of wildfire potential in Portugal during the fire season up to one month in advance based on a statistical approach that integrates information about

meteorological fire danger before and during the fire season. The rationale is that persistent warm and dry conditions along the pre-fire season induce thermal and water stress on vegetation making the landscape more prone to the occurrence of very severe fire episodes and, at the same time, increase the likelihood of heat wave spells that steer the onset and propagation of large fires (Gudmundsson et al., 2014; Turco et al., 2017). The procedure involves three steps; first we set up a null model of burned area (BA) during the fire season (where no meteorological information is incorporated); then we set up a diagnostic

model of BA that incorporates meteorological information during the pre-fire and the fire seasons; finally we set up a prognostic model of BA that only incorporates meteorological information before the fire season. We then show that both the diagnostic and the prognostic models are better performant than the null model and that the loss in performance of the prognostic model (when compared to the diagnostic one) is relatively small, namely in what respects to its capacity to anticipate




fire seasons characterized by high amounts of BA. Section 2 provides a description of data and methods, and results are presented in section 3. Finally, discussion of results and conclusions are drawn in Section 4.

## 2 Data and methods

The area of interest is defined as the territory of Portugal and the study covers the 39-year period from 1980 to 2018. Data of
BA consist of yearly amounts of cumulated BA in July and August, hereby referred to as the fire season. BA data are derived from the official Portuguese Rural Fire Database provided by the National authority for forests (ICNF). The database contains more than half a million records of fire events, with information of total burned area, date and time of ignition and extinction, and spatial location of the starting point. Details about the database are provided in Pereira et al. (2011). Cumulated BA in the fire season accounts for more than 70% of the total burned area in Portugal (Pereira et al., 2013), and more than 80% of extreme
fire days (defined as the top 5% in terms of radiative energy released by wildfires) occur in July and August (DaCamara and Trigo, 2018).

Information about meteorological fire danger consists of daily values of the Daily Severity Index (DSR) covering the months of April to August. DSR is an extension of the Canadian Forest Fire Weather Index System (CFFWIS) and rates the difficulty of controlling fires (van Wagner, 1987). This index has been successfully used to model BA variability in Portugal at daily
and monthly scales (Calado et al. 2008; Pereira et al., 2013). Suited for spatial and temporal averages, DSR results from a direct transformation of the Fire Weather Index (FWI), the last of the six components of CFFWIS. For each day, the six components are computed based on consecutive daily observations of meteorological parameters of the previous days, namely temperature, relative humidity, wind speed, and 24-hour cumulated precipitation (Wang et al., 2015). In this study, meteorological parameters consist of gridded daily values at 12 UTC of 2 meter temperature, relative humidity, wind speed
and 24 hours cumulated precipitation that were obtained from the ERA-Interim reanalysis dataset (Dee et al., 2011) issued by ECMWF. As described in Pinto et al. (2018), the original ERA-Interim data were re-projected onto the normalized geostationary projection (NGP) of Meteosat Second Generation (MSG) (EUMETSAT, 1999), with an average pixel size of about 4 km × 4 km over Portugal. Daily values of DSR for Portugal were then obtained by averaging over all grid points located within the study area.

Following Nunes et al. (2014), the period from April to August is divided into two subperiods; 1) the pre-fire season that runs from April 1 (day 1) up to June 30 (day 91) and 2) the fire season that runs from July 1 (day 1) up to August 31 (day 62).

For each year of the period 1980-2018, meteorological fire danger at day $d$ of the pre-fire season (pfs) of the considered year is rated by index $D_{pfs}(d)$, defined as the cumulative value of daily DSR since April 1:

$$D_{pfs}(d) = \sum_{i=1}^{d} DSR_i, \quad d = 1, \cdots, 91 \tag{1}$$

where $DSR_i$ is the value of DSR at day $i$. It is worth noting that, as $d$ runs along the pre-fire season of the year considered, progressively more information is integrated in $D_{pfs}(d)$ about past daily meteorological fire danger.





In turn, meteorological fire danger of the fire season (fs) of each year of the period 1990-2018 is rated by index $D_{fs}$, defined as the square root of the mean squared anomalies performed over days characterized by a positive anomaly of DSR:

$$D_{fs} = \sqrt{\frac{\sum_{j=1}^{62} H[A_j](A_j)^2}{\sum_{j=1}^{62} H[A_j]}} \tag{2}$$

where $H[x]$ is the Heaviside step function ($H[x] = 1, x > 0; H[x] = 0, x \leq 0$) and:

$$A_j = DSR_j - \overline{DSR_j} \tag{3}$$

is the anomaly of DSR at day $j$ of the considered year, that is defined as the departure of DSR from the climatological mean $\overline{DSR_j}$ for day $j$ (as obtained by averaging DSR for that day over the 39-year period 1980-2018).

As discussed in Pereira et al. (2005), the interannual variability of BA in Portugal is modulated by two kinds of meteorological factors, namely the temperature and precipitation regimes during the pre-fire season and the occurrence of hot and dry spells during the fire season. The former meteorological factor is quantified by $D_{pfs}(d)$, where, for a given day $d$ of the pre-fire season, large values indicate persistent warm and/or dry conditions up to day $d$, inducing thermal and water stress on vegetation and a shortage of water in the soil. The second meteorological factor is in turn quantified by $D_{fs}$ that is very sensitive to the occurrence, during the considered fire season, of very large positive anomalies of DSR that are usually associated to heat waves (Gudmundsson et al., 2014; Turco et al., 2017).

Obtained sets of indices, $D_{pfs}(d)$ and $D_{fs}$, for the respective 39 pre-fire and fire seasons (1980-2018), are normalized by subtracting the respective sample mean and then dividing by the sample standard deviation. Obtained normalized indices will be hereby denoted as $\psi(d)$ and $\chi$, respectively.

The following types of models are considered in this study:

1) A null model $X \sim N(X; \mu_X, \sigma_X)$, where $X$ is the considered variable and $\mu_X$ and $\sigma_X$ are the mean and the standard deviation to be estimated, respectively;

2) A nested model with one covariate $X \sim N(X; p \times A + q, \sigma_X)$ where the mean of the normal distribution linearly depends on covariate A, and $p$ and $q$ are parameters to be estimated;

3) A nested model with two covariates $X \sim N(X; a \times A + b \times B + c, \sigma_X)$ where the mean of the normal distribution linearly depends on covariates A and B, with $a, b$ and $c$ being parameters to be estimated.

Using the maximum likelihood method (Wilks, 2011), estimates of parameters $\mu_X, \sigma_X, p, q, a, b$ and $c$ are obtained as follows from sample $X_j$ $(j = 1, \cdots, n)$ where $n$ is the size of the sample:

$$\hat{\mu}_X = \frac{1}{n} \Sigma(X_j) \tag{4}$$

$$\hat{\sigma}_X^2 = \frac{1}{n} \Sigma(X_j - \hat{\mu}_X)^2 \tag{5}$$





$$p = \frac{cov(x,A)}{var(A)} \tag{6}$$

$$q = \overline{X} - p\overline{A} \tag{7}$$

$$a = \frac{var(B)\ cov(X,A) - cov(A,B)\ cov(X,B)}{var(A)\ var(B) - [cov(A,B)]^2} \tag{8}$$

$$b = \frac{var(A)\ cov(X,B) - cov(A,B)\ cov(X,A)}{var(A)\ var(B) - [cov(A,B)]^2} \tag{9}$$

$$c = \overline{X} - a\overline{A} - b\overline{B}\ , \tag{10}$$

where $\overline{()}$, $var()$ and $cov()$ respectively denote the mean, the variance and the covariance operators.

In the case of the null model, the Lilliefors test (Conover, 1980) is used to test the null hypothesis that the data come from a normal distribution. In the case of the nested models with covariates, the likelihood ratio test (Wilks, 1938) is used to decide on the null hypothesis that the null model is to be retained (against the alternate model under consideration).

**3 Results**

**3.1 Null model**

The time series of cumulated BA in the fire season for the period 1980-2018 (Figure 1) presents very large interannual variability, the extremely high amounts of 2003 and 2005 contrasting with the extremely low values that were observed in 1983, 1988, 1997 and 2008. It is worth referring that 2017, the year with the largest record in total BA (circa 450,000 ha), only

ranks fourth when restricting to July and August because the largest fire events took place out of the fire season, in June and October (Sánchez-Benítez et al. 2018).





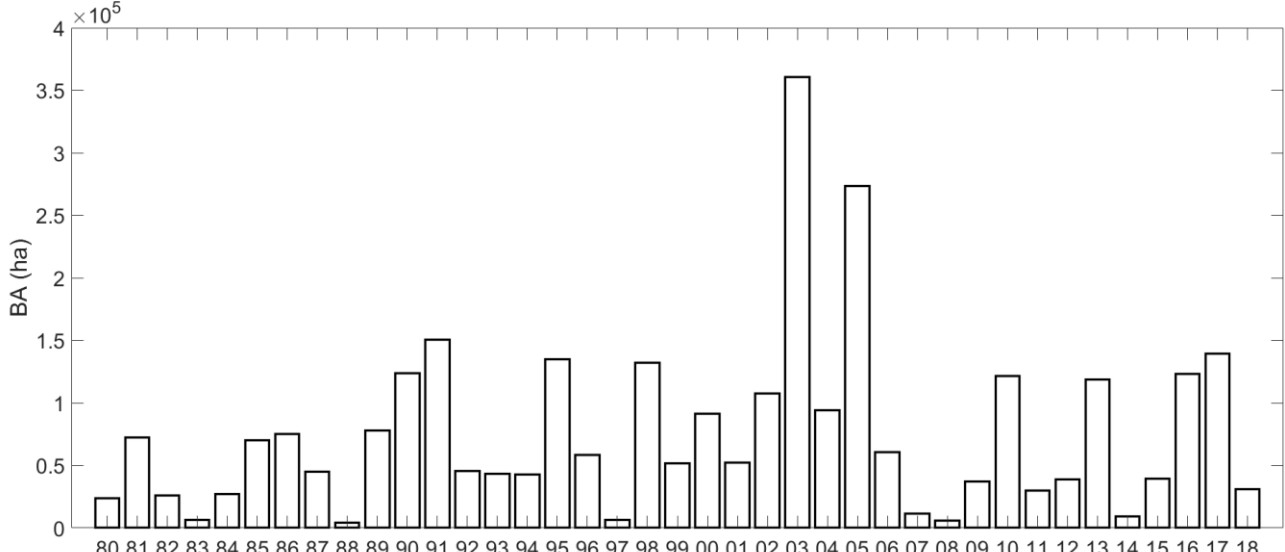

**Figure 1: Time series of yearly cumulated BA (ha) in the fire season (July and August).**

The decimal logarithm of cumulated BA in the fire season follows a normal distribution model (Figure 2); the null hypothesis that the sample of $\log_{10} BA$ comes from a normal distribution is not rejected at the 5% significance level by the Lilliefors test

5 (p-value of 0.21). We have therefore the following null model:

$$\log_{10} BA \sim N(\log_{10} BA; \mu_{BA}, \sigma_{BA}) \qquad (11)$$

with $\mu_{BA} = 4.69$ and $\sigma_{BA} = 0.46$ as obtained by maximum likelihood method.

It is worth noting that values of $\log_{10} BA$ above percentile 80 of the model (highlighted in red in Fig. 2) and below percentile 20 (highlighted in green) present larger departures from the fitted normal distribution than the remaining years; these two groups are classified respectively as severe years (1991, 1995, 1998, 2003, 2005 and 2017) and as weak years (1983, 1988, 1997, 2007, 2008 and 2014). The remaining years (marked in black in Fig.2) are classified as moderate.



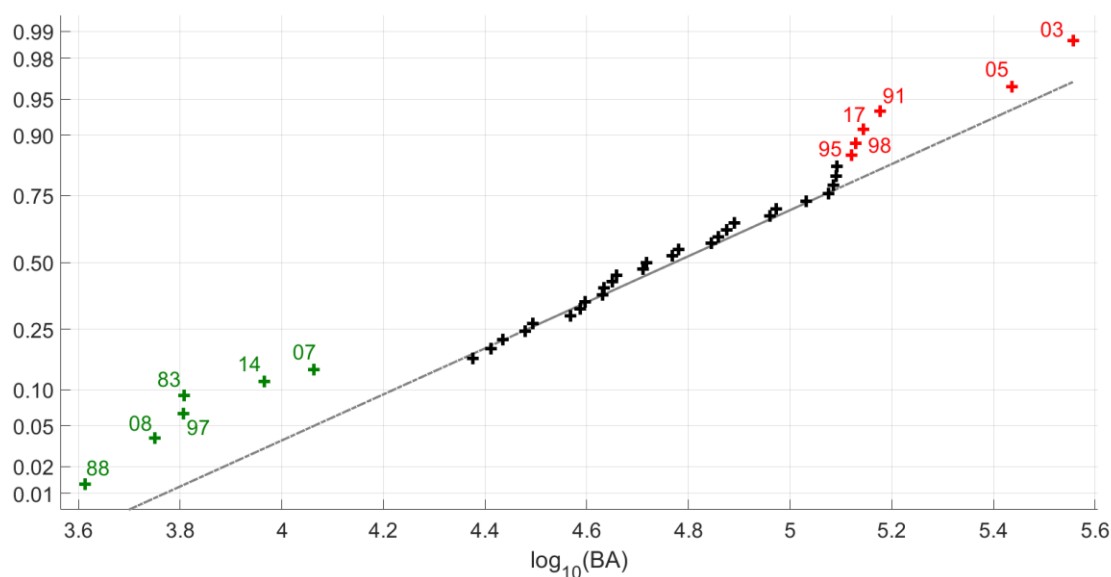

**Figure 2: Normal probability plot comparing the sample of $\log_{10} BA$ to the normal distribution. The groups of severe and weak years are marked in red and green, respectively and the years are identified by the two last digits.**

### 3.2 Model with two covariates

Departures of severe and weak years from the fitted normal distribution suggest that other factors, namely meteorological ones, are playing a role in the interannual variability of BA (Pereira et al., 2005). Following the approach proposed by Nunes et al. (2014), for a given fixed day $d$, where d is chosen between May 26 ($d = 56$) and June 30 ($d = 91$), we tested an alternate model that incorporates information about meteorological fire danger in the pre-fire and the fire seasons. Accordingly, using Equations (1)-(3), we fitted to the sample of $\log_{10} BA$ a normal distribution with the mean linearly depending on covariate

$\psi(d)$ for the considered fixed day $d$, and on covariate $\chi$, i.e., we tested the following model:

$$\log_{10} BA \sim N(\log_{10} BA;\ a(d) \times \psi(d) + b(d) \times \chi + c,\ \sigma_{BA}) \tag{12}$$

It is worth noting that index $d$ is fixed in Equation (12) and is only used to identify the fixed day of the pre-fire season when

covariate $\psi(d)$ is computed. Maximum likelihood estimates of coefficients $a(d)$ and $b(d)$ for the considered fixed day $d$ are then obtained according to Equations (8) and (9). It is also worth noting that parameter $c$ in Equation (12) does not depend on chosen day $d$ since, according to Equation (10), $c = \mu_{BA}$ for all days because $\psi(d)$ and $\chi$ have zero mean (since they are normalized).

A model is accordingly fitted for each fixed day $d$, and, for all days between May 26 ($d = 56$) and June 30 ($d = 91$), it is

found that the null hypothesis that the null model is to be retained (against the alternate nested model) is rejected by the





likelihood ratio test at the 5% significance level, the p-values steeply decreasing from $9 \times 10^{-3}$ for the model fitted on May 26 ($d = 56$) to $6 \times 10^{-4}$ for the model fitted on June 30 ($d = 91$). In turn, when $d$ progresses along the pre-fire season, obtained values of $a(d)$ increase whereas corresponding values of $b(d)$ slightly decrease (Figure 3).

For each day $d$, performance of the fitted alternate model may be assessed by representing each year in space $(\psi, \chi)$ framed
by covariates $\psi(d)$ and $\chi$ over a background of probability of exceedance of a given fixed threshold, e.g. of the mean value $\mu_{BA}$ of the null model, i.e. $P_{exc}(\psi, \chi) = N[\log_{10} BA > \mu_{BA}; a(d) \times \psi(d) + b(d) \times \chi, \sigma_{BA}]$. Results of models fitted on May 26 ($d = 56$), June 15 ($d = 76$) and June 30 ($d = 91$) are shown in Figure 4. It is worth noting that severe (weak) years tend to spread over the upper right (lower left) quadrants of the space indicating that they are associated to high (low) values of both $\psi(d)$ and $\chi$. Severe (weak) years are also associated to high (low) values of $P_{exc}$ and, excepting 2007 in May 26 and June
15, the groups of severe and weak years are fully separated by contour $P_{exc} = 0.5$ (that, in fact, corresponds to the probability of exceedance of $\mu_{BA}$ in the null model, where meteorological factors are not taken into account). Values of $P_{exc}$ associated to severe years tend to gradually increase along the pre-fire season and, on June 30, five out the six severe years present values above 0.7 (a threshold that is barely surpassed by just two of the 27 moderate years). Finally, it is worth noting that the contour lines of $P_{exc}(\psi, \chi)$ become steeper along the pre-fire season, a result in agreement with the steep increase of parameter $a$
(Figure 3).

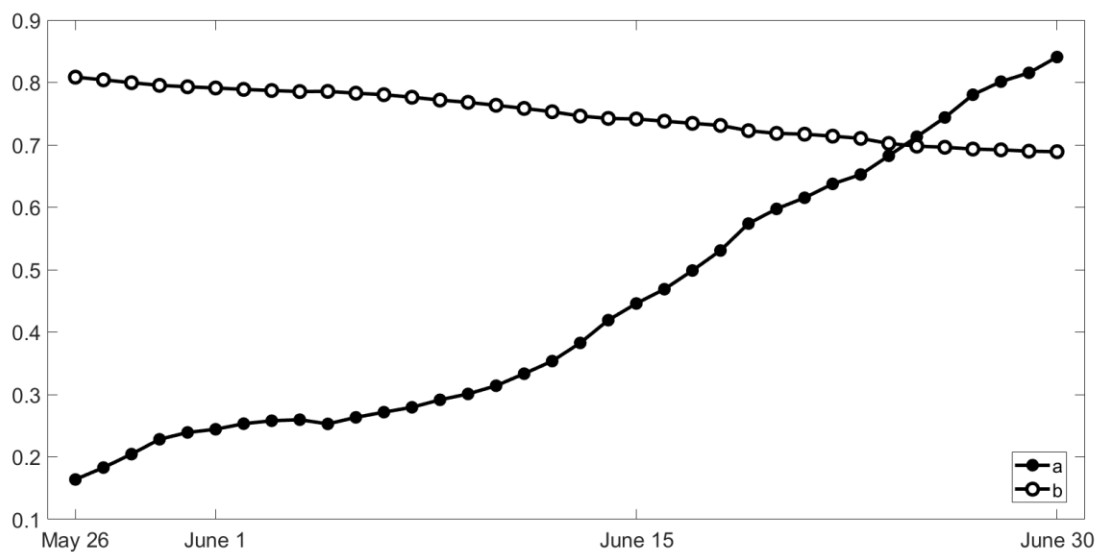

**Figure 3: Temporal evolution of coefficients $a$ and $b$ of the normal model with two covariates.**



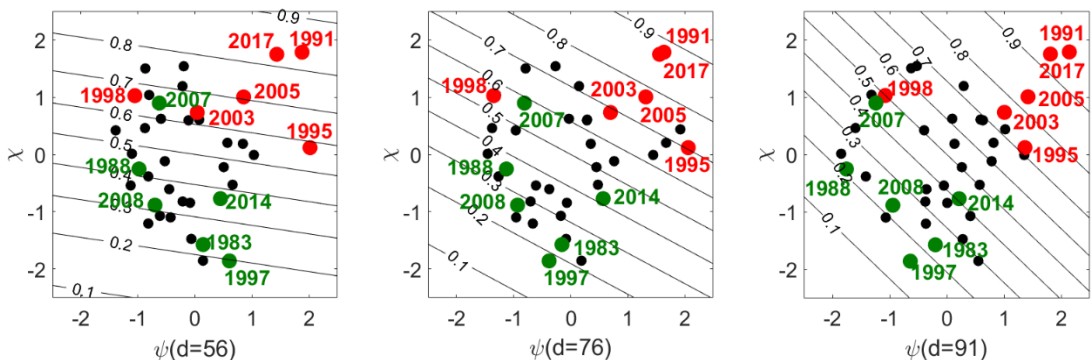

**Figure 4: Scatter plots of the 39-year sample in the space of covariates $\psi(d)$ vs. $\chi$ for May 26 ($d = 56$), June 15 ($d = 76$) and June 30 ($d = 91$). Groups of severe (weak) years are identified by the red (green) circles. The straight lines are contours of $P_{exc}(\psi, \chi)$.**

Results obtained suggest that the likelihood of a given year to belong to the severe or to the weak groups may be estimated based on the value of $P_{exc}(d)$ of each year as estimated by the fitted model on chosen day d (Figure 5). It is worth noting that, for a given year, values of $P_{exc}(d)$ change very slowly from day to day. This is to be expected since 1) the models are all fitted to the same sample of $\log_{10} BA$, 2) covariate $\chi$ is the same in all models and 3) covariate $\psi(d)$ has high serial correlation since, according to Equation (1) index $D_{pfs}(d)$ accumulates values of DSR since April 1 up to the considered day. Therefore,

any decision taken for a given year, based on the respective estimate of $P_{exc}(d)$ from the model fitted on day $d$, is not expected to drastically change on the next few days unless there is an incoming sequence of very high (or very low) daily values of DSR that will considerably change $\psi(d)$ and therefore $P_{exc}(d)$.

The following two types of decision are tested:

- Type A: If $P_{exc} > 0.5$ at day $d$ then the year is not classified as weak (i.e., it is either moderate or severe); otherwise,

if $P_{exc} \leq 0.5$ then the year is not classified as severe (i.e., it is either moderate or weak).

- Type B: If $P_{exc} > 0.7$ at day $d$ then the year is classified as severe.



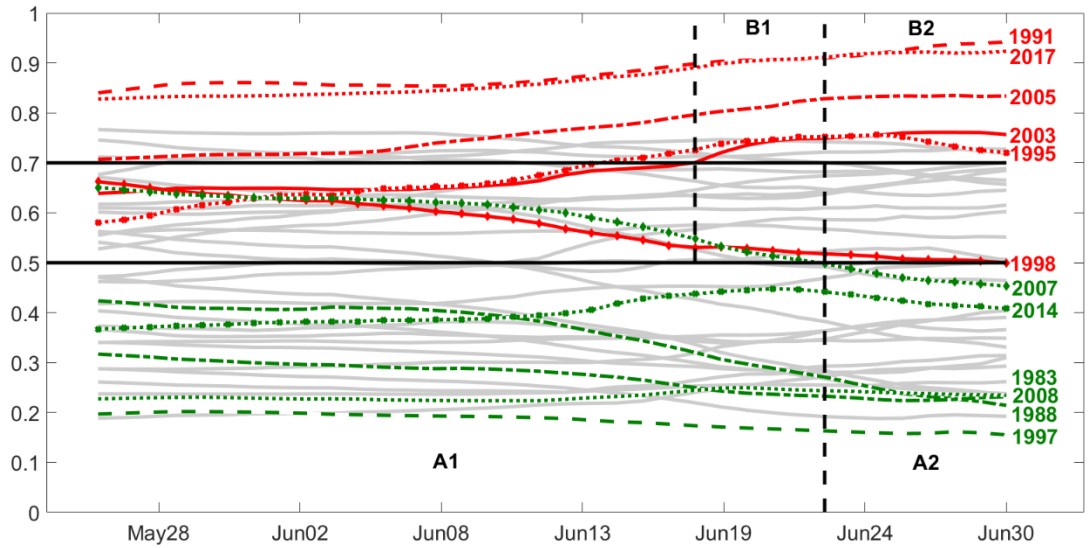

**Figure 5: Temporal evolution of $P_{exc}(\psi, \chi)$ during the pre-fire season (May 26 to June 30). Severe (weak) years are identified by the red (green) curves. The black horizontal lines represent thresholds used in Type A ($P_{exc} = 0.5$) and Type B ($P_{exc} = 0.7$) decisions. Vertical dashed lines delimit the phase where Type A and Type B decisions were checked.**

Results obtained indicate that types A and B of decision should be applied to different periods of the pre-fire season, respectively from May 26 to June 30 (Table 1) and from June 18 to 30 (Table 2). Furthermore, for each type of decision, two phases are identified in the respective periods, the days of each phase being characterized by the same decisions taken for each year. In case of Type A decisions, there is just one wrong decision (2007 is incorrectly classified as not being a weak year)

10 during phase A1 (May 26 to June 22) and all decisions are correct during phase A2 (June 23 to June 30). In case of Type B decisions, and for the entire period (June 18 to 30), all severe years but one (1998) are correctly classified as severe; on the other hand, three moderate years are incorrectly classified as severe during phase B1 (June 18 to 22) and this amount decreases to two during phase B2 (June 23 to 30).

15 **Table 1 Performance assessment of Type A decisions during the pre-fire season (May 26 to June 30) based on the model with two covariates.**

|  | Severe years incorrectly classified as non-severe | Weak years incorrectly classified as non-weak |
|---|---|---|
| **Phase A1** (May 26 – June 22) | None | 2007 |
| **Phase A2** (June 23 - 30) | None | None |



**Table 2 As in Table 1 but for performance assessment of Type B decisions for June 18 to 30.**

|  | Severe years correctly classified as severe | Severe years not classified as severe | Moderate years incorrectly classified as severe |
|---|---|---|---|
| **Phase B1** (June 18 - 22) | 1991, 1995, 2003, 2005, 2017 | 1998 | 1987, 1992, 2010 |
| **Phase B2** (June 23 - 30) | 1991, 1995, 2003, 2005, 2017 | 1998 | 1992, 2010 |

### 3.3 Model with one covariate

Despite its usefulness in characterizing the role played by meteorological factors during the pre-fire and the fire seasons, the model discussed in the last subsection cannot be used to anticipate the likelihood of a given fire season given that one of the two covariates ($\chi$) is derived from daily information during that same fire season. However, results obtained in the previous subsection indicate that the role played by covariate $\psi(d)$ in each fitted model becomes more and more relevant when day $d$ progresses along the pre-fire season as suggested by the steady increase of coefficient $a(d)$ that even becomes larger than

$b(d)$ after June 25 (Figure 3), as well as by the increase in slope of contour lines of $P_{exc}$ (d) (Figure 4).

Therefore, for each considered day $d$, we fitted to the sample of $\log_{10} BA$ the following normal model, now with the mean linearly depending just on covariate $\psi(d)$:

$$\log_{10} BA \sim N(\log_{10} BA; \, p(d) \times \psi(d) + q, \, \sigma_{BA}) \tag{13}$$

Maximum likelihood estimates of model parameters $p(d)$ and $q$ are obtained using Equations (6) and (7) and it may be noted that again parameter $q$ is the same for all fitted model, with $q = \mu_{BA}$ because $\psi(d)$ has zero mean. As in the case of coefficient $a$ in the model with two covariates, coefficient $p(d)$ increases as the considered day $d$ progresses along the pre-fire season (Figure 6). For each day $d$, the relative role played by covariates $\psi(d)$ and $\chi$ may be assessed by comparing the log-likelihood

ratio statistics $-2\ln(L_0/L_1)$ and $-2\ln(L_1/L_2)$, where $L_0$, $L_1$ and $L_2$ are the likelihood functions of the null model, the model with one covariate and the model with two covariates (Figure 7). These two ratios represent the increases in likelihood of the sample of BA when replacing the null model (with no covariates) by the model with covariate $\psi$ and then the latter model by the model with covariates $\psi$ and $\chi$.





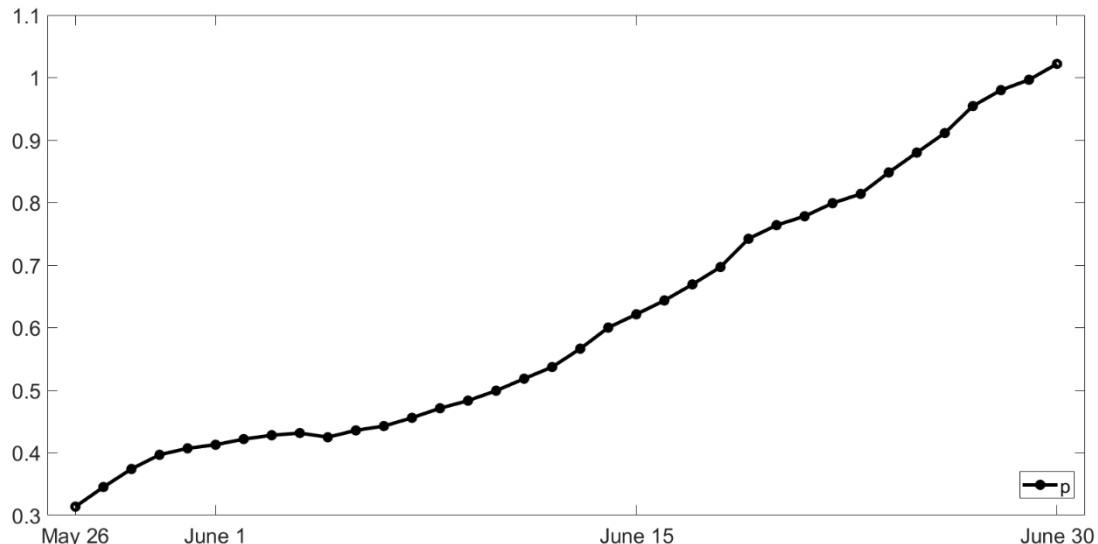

**Figure 6: As in Figure 3 but for coefficient $p$ of the normal model with one covariate.**

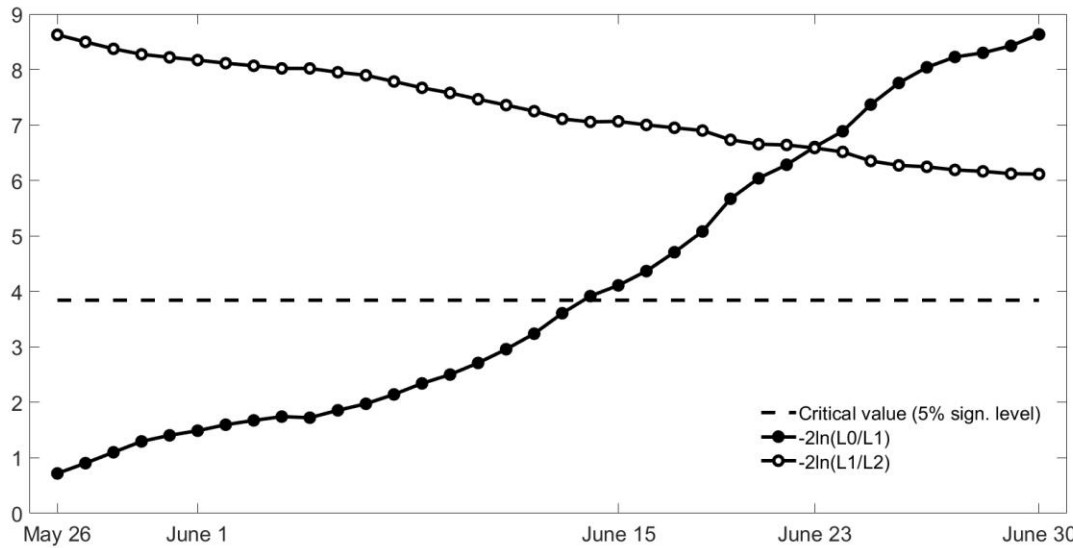

5  **Figure 7: Temporal evolution of the log-likelihood ratio statistics $-2\ln(L_0/L_1)$ between the null model and the model with one covariate and $-2\ln(L_1/L_2)$ between the model with one covariate and the model with two covariates. The horizontal dashed line represents the critical value for the statistic at the 5% significance level.**

Values of $-2\ln(L_0/L_1)$ increase as the considered day $d$ progresses along the pre-fire season, contrasting with the behaviour of $-2\ln(L_1/L_2)$ where a decrease (albeit more moderate) is observed. The former ratio even becomes larger than the latter



from June 23 to 30; however, it may be noted that, for models fitted on days between May 26 and June 13, values of $-2\ln(L_0/L_1)$ are smaller than the critical value at the 5% significance level, indicating that the null model is to be retained (against the alternate model with one covariate). Nevertheless, the model with one covariate is still tested along the entire pre-fire season (May 26 to June 30) because, as shown in Figure 4 (left and central panels), the severe (weak) groups tend to present for the most part high (low) values of covariate $\psi$. Reinforcing the relevance of this covariate when estimating the likelihood of a given year to belong to the severe class, we find that, when, for each considered fixed day $d$, we restrict to years with $\psi(d)$ larger than the respective daily median, there is a positive correlation between $\psi(d)$ and $\chi$ that, except from June 5 to 12, is significant at the 5% level (Figure 8).

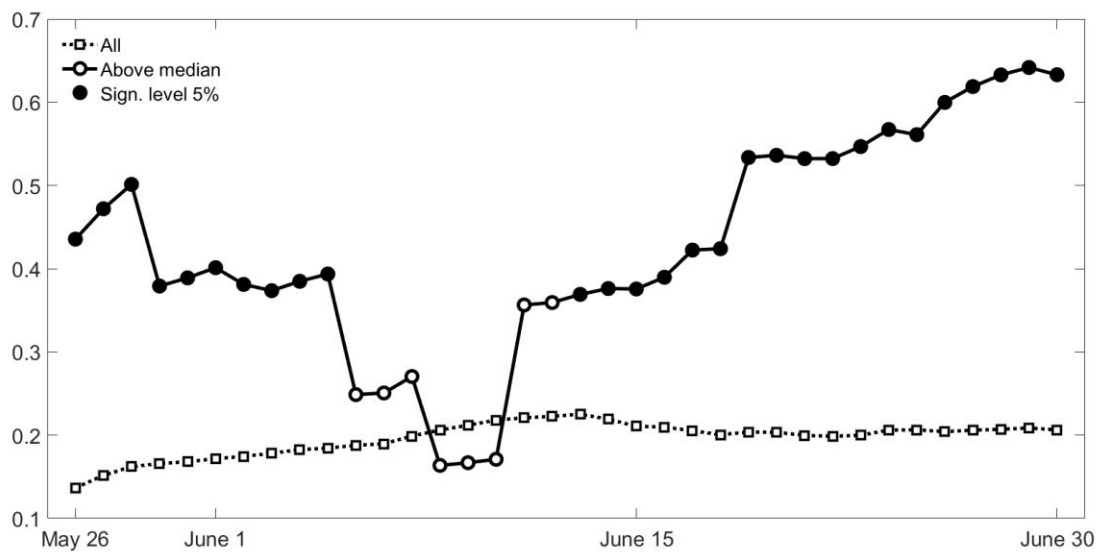

**Figure 8: Temporal evolution of the correlation between $\psi(d)$ and $\chi$ from May 26 to June 30 for all years (dotted curve) and restricting to years $\psi(d)$ larger than the respective median (solid curve). Black circles in the solid curve identify values of correlation that are significant at the 5% level.**

As with the model with two covariates, a given year is anticipated as belonging to the severe or the weak groups by making Type A and Type B decisions based on the value of $P_{exc}(d)$ of each year as estimated by the fitted model (with one covariate) on that fixed day d. It is worth noting the use of the wording "is anticipated as" (instead of "is classified as" employed in the previous section) that is meant to enhance the prognostic character of the model with covariate $\psi$. It is also worth noting that, since the model with one covariate has lower variability than the model with two covariates, the threshold of 0.7 (previously used in Type B decisions) is now lowered to 0.66 (Figure 9).





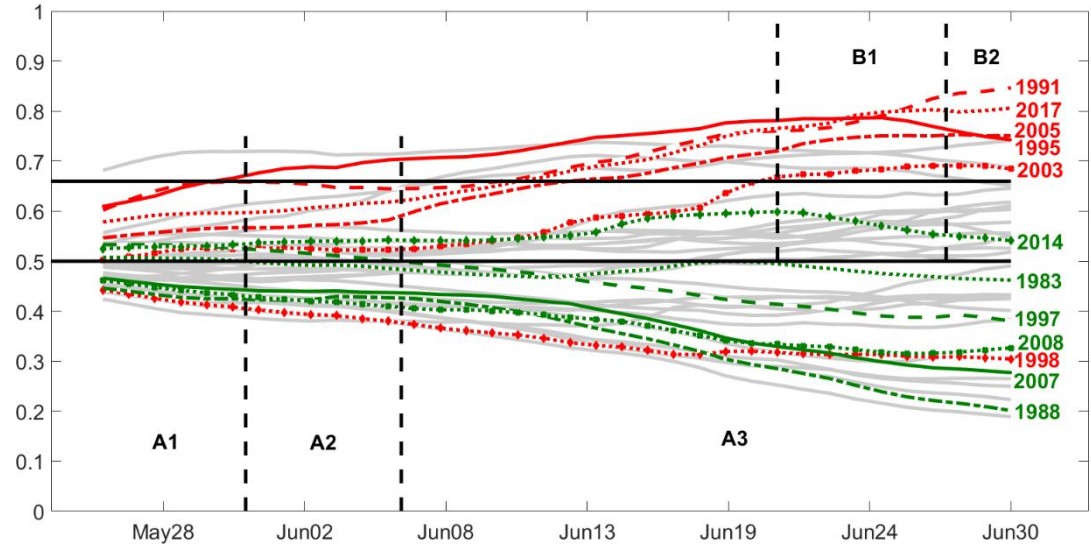

**Figure 9: As in Figure 5 but when using the model with one covariate.**

Three phases (A1, A2 and A3) are used to assess Type A decisions (Table 3). In all phases, only 1998 (out of the six severe

5    years) is incorrectly anticipated as a non-severe year. In turn, the number of weak years incorrectly anticipated as non-weak

decreases from three in phase A1 (May 26 to 30) to two in phase A2 (May 31 to June 5) and then to just one in phase A3 (June

6 to 30). Assessment of Type B decisions (Table 4) is performed in two phases (B1 and B2). Regarding the severe years

correctly classified as severe, it is worth pointing out that, apart from phase B1 beginning three days later (on June 21 instead

of 18), there is no decrease in performance from the model with two covariates, 1998 being again the only missed severe year

10   in phases B1 and B2. Finally, no virtual decrease is found in performance in the number of moderate years incorrectly

anticipated as severe, that, as with the model with two covariates, decrease from three in phase B1 (June 21 to 26) to two in

phase B2 (June 27 to 30).

**Table 3 As in Table 1 but for the model with one covariate.**

|  | Severe years incorrectly anticipated as non-severe | Weak years incorrectly anticipated as non-weak |
|---|---|---|
| **Phase A1** (May 26 – 30) | 1998 | 1983, 1997, 2014 |
| **Phase A2** (May 31 – June 5) | 1998 | 1997, 2014 |
| **Phase A3** (June 6 - 30) | 1998 | 2014 |



**Table 4 As in Table 2 but for the model with one covariate.**

| | Severe years correctly anticipated as severe | Severe years not anticipated as severe | Moderate years incorrectly anticipated as severe |
|---|---|---|---|
| **Phase B1** (June 21 - 26) | 1991, 1995, 2003, 2005, 2017 | 1998 | 1992, 2006, 2015 |
| **Phase B2** (June 27 - 30) | 1991, 1995, 2003, 2005, 2017 | 1998 | 1992, 2015 |

## 4 Discussion and conclusions

The increasing dimension of the impacts of extremely large wildfires that have been affecting Portugal in the last decades calls
for the improvement of diagnostic and prognostic tools designed to assist forest and fire managers in taking better decisions
on both prevention and combat. For this purpose, we set up a model that allows making outlooks, up to one month ahead, of
wildfire potential in Portugal during the fire season (defined as July and August). We started by setting up a null model of BA
consisting of a simple normal distribution fitted to the sample (for the period 1980-2018) of the decimal logarithm of the yearly

cumulated values of BA during the fire season ($\log_{10} BA$). The rationale is in line with previous studies (Pereira et al., 2013;
Nunes et al., 2014) that used the log-normality of monthly cumulated values of BA to model the inter-annual variability of
BA.

The null model was then used to define two groups of years, the severe and weak groups, formed by the years with values
respectively above percentile 80 and below percentile 20 of the fitted normal distribution $\log_{10} BA \sim N(\log_{10} BA; \mu_{BA}, \sigma_{BA})$

(Figure 2). The severe and weak groups presented noticeable deviations from the fitted normal model, an indication that
meteorological factors are likely to be contributing to exacerbate or mitigate the occurrence of large fire events. Indeed, as
shown by Pereira et al. (2005), meteorological conditions preceding the fire season and/or concurring with the fire events in
Portugal explain two thirds of the observed interannual variability of BA during July and August. For instance, long periods
of warm and dry weather during the pre-fire season induce high levels of vegetation stress that increase the probability of

occurrence of large fire events as shown for Portugal (Trigo et al., 2006) and even the Mediterranean basin (Gudmundsson et
al., 2014; Turco et al., 2017). On the other hand, the occurrence during the fire season of extremely hot and dry spells associated
to strong winds is the key triggering mechanism for the onset and spreading of very large fire events (Amraoui et al, 2013).

The role played by meteorological factors was then included in the model of BA by fitting to the sample of $\log_{10} BA$ an
alternate normal model where the mean linearly depends on two covariates, $\psi$ and $\chi$, designed to rate the role played by

meteorological factors respectively during the pre-fire (May 26 to June 30) and the fire seasons (Nunes et al., 2014). Covariate
$\psi$ (defined at a given fixed day of the pre-fire season) responds to the accumulation of thermal and water stress up to the
chosen day, whereas covariate $\chi$ is sensitive to the occurrence of hot and dry spells taking place in the fire season.



The relative magnitudes of the daily coefficients of the two covariates for each chosen day (Equation 12) change along the pre-fire season, with coefficient $b$ of $\chi$ slightly decreasing and coefficient $a$ of $\psi$ increasing and even surpassing $b$ after June 24 (Figure 3). This result reflects the key role played by the accumulation of thermal and water stress of vegetation along the pre-fire season on the occurrence of large fire events during the fire season, a result in line with previous studies focusing on

Portugal (Pereira et al., 2005; Trigo et al., 2006; Calado et al., 2008). Results from the model with two covariates (Figure 4) show that, during phase A2 of the pre-fire season (June 23 to 30), the two groups of severe and weak years are fully separated by the line $P_{exc} = 0.5$, where $P_{exc}(\psi, \chi)$ is the probability of exceedance of $\mu_{BA}$ and that, during phase A1 (May 26 to June 22), the same threshold in probability separates all severe and weak years but 2007 (a weak year). These results stem from the fact that severe (weak) years are characterized for their most part by high (low) values of both covariates, therefore occupying

the upper right (lower left) quadrants in the space $(\psi, \chi)$ of covariates.

Performance of the model with two covariates was evaluated by testing two types of decisions based on the temporal evolution of $P_{exc}$; for Type A, the decision is on whether a given year is classified as not being severe or as not being weak, whereas for Type B it is on whether the year is classified as severe. Type A decisions are made during the entire pre-fire season (May 26 to June 30) and Type B decisions restrict to the end of the pre-fire season (June 18 to 30) when the role played by the cumulation

of vegetation stress becomes as relevant as the one by the occurrence of extreme weather events during the fire season.

As a result of the separability all Type A decisions are correct during phase A2 and only 2007 is classified as non-weak during phase A1. Regarding Type B decisions, out of the six severe years, only 1998 is not classified as severe and this incorrect anticipation results from the fact that 1998 is the only case where a low value of $\psi$ along the pre-fire season is not associated to a low or moderate value of $\chi$.

The model with two covariates has the disadvantage that it cannot be used as a tool to make outlooks about the likelihood of a given year to be severe or weak because covariate $\chi$ is derived from daily information respecting to the fire season itself. However, the fact that severe and weak years tend to concentrate in opposite regions of the $(\psi, \chi)$ space suggests that large values of $\chi$ are more likely to occur after large values of $\psi$, a result that is in line with previous findings that soil moisture deficit and drought (associated to increasingly large values of $\psi$) have an impact on summer hot extremes in Mediterranean

regions (Vautard et al., 2007; Hirschi et al., 2011), a feature that fostered the development of both statistical and dynamical seasonal fire forecasting approaches in the Mediterranean region (Gudmundsson et al., 2014; Turco et al., 2017; Turco et al., 2018). This relationship between $\psi$ and $\chi$ for larger values of $\psi$ further translates into the positive correlation between the two variables for $\psi$ larger than the median during the pre-fire season (Figure 8).

A model with predicting capacity was therefore tested by fitting to the sample of $\log_{10} BA$ an alternate normal model where

the mean linearly depends only on covariate $\psi$ as evaluated for a given fixed day. Performance of the model was again assessed by testing Type A and Type B decisions. In case of Type A decisions, only 1998 (out of the six severe years) is incorrectly anticipated as non-severe. For Type B decisions, results are virtually the same as with the model with two covariates, reflecting the prominent role of cumulated vegetation stress at the end of the pre-fire season in favouring the ocurrence of severe years.



However, a test similar to Type B, but applied to weak years, has poor performance because the quadrant of low values of both $\psi$ and $\chi$ in the space of covariates is occupied by weak years as well as by some of the moderate ones (Figure 4). This may be viewed as an indication that low values of $\psi$ along the pre-fire season have a moderate impact on the likelihood of a following fire season with a low value of $\chi$ and therefore in the likelihood of having a weak year.

Finally it is worth noting that we have not incorporated in the prognostic model (with one covariate) any information derived from the fact that covariates $\psi$ and $\chi$ are positively correlated (when $\psi$ is larger than the median). This would imply setting up a model of the distribution of $\chi$ using information about meteorological conditions along the pre-fire season and then incorporting this information in the prognostic model. This is however beyound the scope of this paper.

A beta version of the proposed model has been experimentally running since May 2017 and the correctness of the outlooks
(made in real time) about 2017 being a severe year (based on a Type B decision) and about 2018 not being severe (based on a Type A decision) give confidence about the potential of the model to be operationally used to anticipate the occurrence of severe years.

Daily outlooks are currently available at CeaseFire ([http://idlcc.fc.ul.pt/CeaseFire/index.php](http://idlcc.fc.ul.pt/CeaseFire/index.php)), a website designed to integrate and disseminate relevant meteorological information to the user fire community by means of a simple, fast and user-friendly
interface (Evans, 2018). Developed by Instituto Dom Luiz (IDL) at the Faculty of Sciences of the University of Lisbon (Portugal), the platform relies on data provided in near real-time by LSA SAF, the EUMETSAT Satellite Application Facility for Land Surface Analysis (Trigo et al., 2011). Currently there are about 900 registered users in the CeaseFire platform, most of them from national authorities and services, as well as from municipalities and private companies, namely from the paper and pulp industry.

**Acknowledgements**

Research was developed within the framework of the EUMETSAT Satellite Application Facility for Land Surface Analysis (LSA SAF) and of FCT project "Forecasting fire probability and characteristics for a habitable pyroenvironment (FireCast)" under grant PCIF/GRF/0204/2017. Part of this work was developed under a contract with *The Navigator Company*.

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
