# Peer review of "Wildland fire potential outlooks for Portugal using meteorological indices of fire danger"

_Natural Hazards and Earth System Sciences, 2019_

## Referee Comment (RC1) · Anonymous Referee #1 · 6 May 2019

The entitled paper "Wildland fire potential outlooks for Portugal using meteorological indices of fire danger" elaborates and compares models using one or two meteorological factors derived from a meteorological daily fire danger index (DSR). The aim of the paper fits with NHESS journal scope and objectives, presenting a useful tool for fire prevention issues. Methods are clearly outlined, I suggest the authors minor changes related to the order of the model description in the methods section (see section specific comments below). The results support the main hypothesis, which lead to reasonable conclusions. Data and methods are proper described, as well as mathematical methods and test applied. In my opinion it might be reproducible for other case studies. The title and abstract reflect the work done and the obtained results. The abstract is understable for a diversified audience. Figures and tables have a proper

size and resolution. Regarding some of the captions of the figures please see the section technical corrections below. The authors provide the previous related works following them and contributing with some specific pre fire season information. The number of references is adequate and can be found. Generally speaking the paper is well structured and with a proper length.

Specific comments Data and methods: - Why only July and August as fire season? As you mentioned in the results section in the year 2017 the highest BA values occurred out of this period - Even tough there is the reference to the explanation about the Portuguese Rural Fire database, what is the reference unit of the BA? Perimeters? Total BA at municipality or parish level?

Results: - In the data and methods section you explain the models considered as 1) null model, 2) a nested model with one covariate and 3) a nested model with to covariates. Then, in the results section, the model with two covariates is coming first (3.2) than the one with one covariate (3.3). As 3.3 section is explained/justified of what happened in the previous section, maybe you could consider to change the order in the methods section. - In page 13 you mention that the threshold of 0.7 is lowered to 0.66. Why 0.66?

Discussion and conclusions: Are there any future developments projected of these models incorporating any other kind of covariates, not only meteorological derived? Is the performance of those models being influenced of any possible missing covariates?

Technical corrections Page 1, line 25: of2017 Page 2, line 18: System(EFFIS) Page 3, line 26: July 1st , August 31st Page 3, line 28: April 1st Page 4, line 1: 1990-2018 is Page 5, line 7: Lilliefors test Page 7, line 20: please revise the sentence Page 8, line 4: performance of Page 9, figure 4: please add a), b) c) if it is according to the journal graphic rules Page 12, figure 6: the caption should be self explicative, no need to go to check a previous figure. Please rewrite Page 14, figure 9: same comment as before You use "worth noting" many times in the text.

СЗ

---

## Referee Comment (RC2) · Anonymous Referee #2 · 9 May 2019

I read with the interest the manuscript by Nunes et al., the authors use a rather simple and sound methodology to forecast the intensity of the fire season in Portuga. the main idea is that a large part of the variability in summer burnt areas in Portugal could be explained (and therefore predicted) by the prefire season spring conditions. There are two raisons for that. First, fire activity is largely driven by fuel moisture conditions, which are in turn dependent on meteorological spring conditions. Second, it has been shown that dry spring conditions increase the likelihood of heat waves during summer, which are known to be associated with large fire occurrence in Portugal. The performance and degree of predictability of their modeling framework is assessed by considering whether the model can discriminate weak and severe years and at which date can this model provide robust predictions. The authors conclude that, given the relatively good

performance (severe or weak years can be predicted up to one month in advance), this approach can be of interest for the fire community.

This is an interesting work that can have operational applications for fire management/suppression in Portugal and also stimulate the application and developments of some similar approaches in other regions. The method and result sections are clear and well written, with some interesting analyses (such as the evaluation of type A and B decisions for instance). I have a series of recommendation to improve this paper even more.

Overall, I found the discussion was rather descriptive, mostly rewording the main results and methods of the study, without bringing much information and developments. The main focus on theoretical/fundamental aspects related to the fire-weather relationships while this is not the objective of the paper. One could expect that the discussion would bring much information about the added value of the work comparing to other methods (does that really bring much information than seasonal forecasts ?), its potential use for direct operational applications, the potential improvements (other indices, why is the year 1998 not correctly classified?)...

I was also surprised by the choice of your indices for the prefire season (Dpfs). I d'ont see the rationale for using the cumulated DSR as an index for "vegetation stress". First the DSR includes many other information that is not related to vegetation dryness (such as wind speed). Second, as the DSR already depends on its previous values, I don't understand why you should use its cumulative value. The way I see it, a daily index that is recognized to be a proxy of vegetation dryness (e.g. the Drought code of the FWI among others) would be more appropriated here. Also, and If your objective was to obtain the best performance I was also wondering why you did not compare several indices.

Finally, I think that much emphasis is placed on the "diagnostic model of BA", i.e. using meteorological information during the pre-fire and the fire seasons while this model

does not bring much added value to the field.

Other comments

P2, L25-28: That part is interesting and should be developed. You should also provide some citations that shows the links between spring drought and the likelihood of summer heatwaves. Not sure the references are appropriate

P3, L15: could you provide more details on how does the DSR differs from FWI.

P4 : L8-14: That would be clearer if that part was moved before indices descriptions

P4 : L16, is psi(d) computed according to the mean and standard deviation of day d or over the entire population (including all days)

P7, L7, why starting from May 26 only. That would be interesting to start earlier to see also when does that information becomes relevant.

---

## Author Comment (AC1) · 24 Jun 2019

The authors thank Reviewer #1 for his positive opinion about the paper.

All changes made in the manuscript are marked in yellow.

Specific comments:

- Data and methods:

Question: Why only July and August as fire season? As you mentioned in the results section in the year 2017 the highest BA values occurred out of this period. Answer: As stated in the Data and Methods section, cumulated burned area in the fire season accounts for more than 70% of the total burned area in Portugal, and more than 80% of

extreme fire days occur in July and August. Outlooks for July and August are therefore especially relevant when planning the fire season. Moreover, when we attempted to include in the fire season the burned area for the month of September, the overall quality of the model decreased, most likely because fire events during September are less dependable on spring conditions. However, as pointed out in several studies and reports, the length of the fire season has been increasing, and the atypical 2017 fire season (with extremely large fire events in June and October) is consistent with this trend. This issue is now addressed in the Discussion and conclusions section.

Question: Even though there is the reference to the explanation about the Portuguese Rural Fire database, what is the reference unit of the BA? Perimeters? Total BA at municipality or parish level? Answer: For each year, the total amount of BA data during the fire season was estimated by adding the contributions of all recorded events as obtained from the official Portuguese Rural Fire Database provided by the National authority for forests (ICNF). This is now clarified in the manuscript.

- Results

Question: In the data and methods section you explain the models considered as 1) null model, 2) a nested model with one covariate and 3) a nested model with to covariates. Then, in the results section, the model with two covariates is coming first (3.2) than the one with one covariate (3.3). As 3.3 section is explained/justified of what happened in the previous section, maybe you could consider to change the order in the methods section. Answer: We acknowledge the suggestion raised by the reviewer and therefore the order of presentation of the models was reversed in the Data and Methods section.

Question: In page 13 you mention that the threshold of 0.7 is lowered to 0.66. Why 0.66? Answer: As stated in the manuscript, the model with only one covariate, psi(d), presents lower variability than the model with covariates psi(d) and chi. Accordingly, the threshold was lowered to 0.66, a benchmark that corresponds to the upper tercile of the distribution.

- Discussion

Question: Are there any future developments projected of these models incorporating any other kind of covariates, not only meteorological derived? Is the performance of those models being influenced of any possible missing covariates? Answer: The discussion section was entirely rewritten and all the points raised by the reviewer here are now addressed.

Technical corrections

Page 1, line 25: of2017 Page 2, line 18: System(EFFIS) Page 5, line 7: Lilliefors test Page 8, line4: performance of These typos were not in the original word document. They are related to the conversion from Word to pdf during the submission process.

Page 4, line 1: 1990-2018 is The typo was corrected and now the text reads "1980-2018 is".

Page 3, line 26: July 1st, August 31st Page 3, line 28: April 1st All dates in the manuscript were converted to the proposed format.

Page 7, line 20: please revise the sentence The sentence was changed.

Page 9, figure 4: please add a), b) c) if it is according to the journal graphic rules Figures were changed as suggested.

Page 12, figure 6: the caption should be self explicative, no need to go to check a previous figure. Please rewrite Page 14, figure 9: same comment as before Captions were changed as suggested.

You use "worth noting" many times in the text. The expression "worth noting" was removed when possible and now appears just four times in the manuscript.

Please also note the supplement to this comment:
https://www.nat-hazards-earth-syst-sci-discuss.net/nhess-2019-60/nhess-2019-60-

AC1-supplement.pdf

[Figure]

**Supplement:**

[revised manuscript text omitted]

The model consists of a normal distribution of the decimal logarithm of the yearly cumulated values of BA during the fire season ($\log_{10} BA$) where the mean of the distribution linearly depends on a covariate $\psi(d)$, defined at a given fixed day $d$ of the pre-fire season, which responds to the accumulation of thermal and water stress up to the chosen day $d$. This model with one covariate results from the simplification of a model with two covariates, respectively $\psi(d)$ and $\chi$, the latter covariate being sensitive to the occurrence of hot and dry spells associated to strong winds that are the key triggering mechanism for the onset and spreading of very large fire events (Amraoui et al, 2013).The rationale behind setting up the model with covariate $\psi(d)$ is twofold: 1) long periods of warm and dry weather during the pre-fire season induce high levels of vegetation stress that increase the probability of occurrence of large fire events as shown for Portugal (Trigo et al., 2006) and even the Mediterranean basin (Gudmundsson et al., 2014; Turco et al., 2017); 2). In fact, it is now well-known that soil moisture deficit and drought (associated to large values of $\psi$) have an impact on summer hot extremes in Mediterranean regions (Vautard et al., 2007; Hirschi et al., 2011), a feature that fostered the development of both statistical and dynamical seasonal fire forecasting approaches in the Mediterranean region (Gudmundsson et al., 2014; Turco et al., 2017; Turco et al., 2018).

Performance of the model was evaluated for the 39-year period 1980-2018 by testing two types of decisions based on the temporal evolution of $P_{exc}$; for Type A, the decision is on whether a given year is classified as not being severe or as not being weak, whereas for Type B it is on whether the year is classified as severe. Type A decisions are made during the entire pre-fire season (May 26$^{th}$ to June 30$^{th}$) and Type B decisions restrict to the end of the pre-fire season (June 21$^{st}$ to 30$^{th}$) when the role played by the cumulation of vegetation stress becomes as relevant as the one by the occurrence of extreme weather events during the fire season. In case of Type A decisions, only 1998 (out of the six severe years) is incorrectly anticipated as non-severe. For Type B decisions, all severe years but 1998 were correctly antecipated as severe, reflecting the prominent role of

cumulated vegetation stress at the end of the pre-fire season in favouring the ocurrence of severe years. However, a test similar to Type B, but applied to weak years, has poor performance and this may be viewed as an indication that low values of ψ along the pre-fire season have a moderate impact on the likelihood of a following fire season with a low value of χ and therefore in the likelihood of having a weak year.

5    Since both Type A and Type B decisions have shown to be incorrect for the fire season of 1998, the temporal evolution of daily values of DSR is worth being analysed along the pre-fire and the fire seasons (Figure 10). Excepting for two peaks in the last 10 days of June, the pre-fire season of 1998 is dominated by moderate daily values of DSR which are very close to zero between May 10th and early June, a period characterized by significant amounts of rainfall. Therefore, by the end of May, vegetation was not subject to either thermal or water stress and this translates into low values of $\psi(d)$ along the pre-fire season

10   (Figure 4). This unstressed condition of vegetation explains why the two large peaks of daily DSR in the second half of June did not trigger any large fire events as well as why the large peak of DSR at the end of the first half of July was followed by days of moderate burned area. However, the meteorological conditions drastically changed in August, with the first half of the month being dominated by values of daily DSR well above the median that culminated with a sequence of days with DSR much higher than percentile 90, and the second half presenting a peak of DSR very close to percentile 90. The impact of this

15   high number of extreme days translates into a large value of χ for 1998, the 7th largest in the 39 years analysed and the third of the six severe years. Striking a vegetation stressed by the warm and dry conditions that prevailed since the beginning of August, the two peaks of DSR triggered two sequences of days affected by large fire events that made 1998 rank 5th in cumulated BA in 1980-2018.

[Figure]

20   **Figure 10: Temporal evolution of daily values of DSR for 1998 (red curve) and of the daily values of the median (solid black curve) and of percentile 90 (dotted black curve) for the period 1980-2018. The dark grey bars indicate the cumulated daily values of BA (in hectares) for Portugal. The vertical black line subdivides the period into pre-fire and fire seasons.**

As shown in Figure 4, 1998 is the only severe year where a value of $\psi$ at the last day of the pre-fire season (June 30[th]) below the mean minus one standard deviation is followed by a value of $\chi$ above the mean plus one standard deviation. The exceptional character of 1998 is reinforced by the fact that, even when considering all 39 years, this condition is fulfilled in only 2 years, i.e. 1998 and 1985 (a moderate year). However, despite the extreme meteorological conditions observed in August, 1998 just ranks 5[th] among the six severe years, suggesting that the mild conditions during the pre-fire season still had a mitigating role. On the other hand, the exceptionality of 1998 puts an emphasis on the importance of short-term extreme events as triggers of large-fire events, a feature that is not considered when restricting to the information provided by maps of long-term seasonal forecasts of temperature and rainfall anomalies, undermining the usefulness of current seasonal forecasting outputs for wildland fire potential outlooks.

Several improvements to the proposed prognostic model (with covariate $\psi$) are currently being considered. First, no advantage was taken of the fact that covariates $\psi$ and $\chi$ are positively correlated (when $\psi$ is larger than the median). This information may be added by setting up a model of the distribution of $\chi$ using information about meteorological conditions along the pre-fire season and then incorporting this information in the prognostic model. Second, the model was developed for the entire territory of Portugal, not taking into account regional characteristics of climate, land cover and fire regime. It is therefore worth setting up regional models over areas with distinct pyrogeographical characteristics. Finally, the outlooks respect to the months of July and August and given the observed trend to have a longer fire season it would be very useful to extend them until October. For instance, as pointed out by Beighley and Hyde (2018), 36% of the total area burned in 2009-2017 was outside the period from July to September, a fraction that is three times larger than the value of 12% observed when considering the period 2001-2008.

A beta version of the proposed model has been experimentally running since May 2017 and the correctness of the outlooks (made in real time) about 2017 being a severe year (based on a Type B decision) and about 2018 not being severe (based on a Type A decision) give confidence about the potential of the model to be operationally used to anticipate the occurrence of severe years.

Daily outlooks are currently available at CeaseFire (http://idlcc.fc.ul.pt/CeaseFire/index.php), a website designed to integrate and disseminate relevant meteorological information to the user fire community by means of a simple, fast and user-friendly interface (Evans, 2018). Developed by Instituto Dom Luiz (IDL) at the Faculty of Sciences of the University of Lisbon (Portugal), the platform relies on data provided in near real-time by LSA SAF, the EUMETSAT Satellite Application Facility for Land Surface Analysis (Trigo et al., 2011). Currently there are about 900 registered users in the CeaseFire platform, most of them from national authorities and services, as well as from municipalities and private companies, namely from the paper and pulp industry.

**Acknowledgements**

Research was developed within the framework of the EUMETSAT Satellite Application Facility for Land Surface Analysis (LSA SAF) and of FCT project "Forecasting fire probability and characteristics for a habitable pyroenvironment (FireCast)" under grant PCIF/GRF/0204/2017. Part of this work was developed under a contract with *The Navigator Company*.

[revised manuscript text omitted]

---

## Author Comment (AC2) · 24 Jun 2019

The authors thank Referee #2 for his very constructive comments.

All changes made in the manuscript are marked in yellow.

Question: One could expect that the discussion would bring much information about the added value of the work comparing to other methods (does that really bring much information than seasonal forecasts?), its potential use for direct operational applications, the potential improvements (other indices, why is the year 1998 not correctly classified?)...

Answer: The Discussion section was entirely rewritten. The case of 1998 is now addressed in detail, the added value of the proposed model when compared with seasonal forecasts is discussed and improvements of the model are considered.

Question: I was also surprised by the choice of your indices for the pre fire season (Dpfs). I don't see the rationale for using the cumulated DSR as an index for "vegetation stress". First the DSR includes many other information that is not related to vegetation dryness (such as wind speed). Second, as the DSR already depends on its previous values, I don't understand why you should use its cumulative value. The way I see it, a daily index that is recognized to be a proxy of vegetation dryness (e.g. the Drought code of the FWI among others) would be more appropriated here. Also, and If your objective was to obtain the best performance I was also wondering why you did not compare several indices.

Answer: The Drought code (DC) is sensitive to the slow-varying conditions of soil moisture in deep compact duff layers. DC is mainly controlled by precipitation and this slow-acting moisture code is especially useful as a warning when the lower layers of duff may be drier than the upper ones (van Wagner, 1987). However, the slow rate of change of DC makes it too insensitive to meteorological changes that may change soil moisture at the intermediate and surface layers. FWI, in turn, presents the advantage of reflecting the conditions of soil moisture at deep, intermediate and surface layers; FWI was nevertheless designed to be used at the daily level and is too sensitive to day to day changes in meteorological conditions. A compromise may be achieved by adding up the daily contributions of FWI. However, the high peaks of FWI introduce distortions in the cumulated values, but this problem is mitigated when using DSR that is obtained from FWI by means of a transformation that weights FWI sharply as it raises. In fact, there is a long tradition in Portugal to use cumulated values of DSR (since January 1st of each year) as an indicator of proneness of vegetation to burn. Moreover, as stated in the manuscript, cumulated DSR was successfully used as a meteorological predictor in previous studies. A first assessment of the potential of DC and of cumulated DSR to discriminate between severe and weak years is provided in Fig.1 and 2 that presents the temporal evolution of the medians of DC (upper panel) and cumulated DSR (lower

panel) for all 39 year of 1980-2018 (black curve) and for the subsets of severe (red curve) and weak (green curve) years. Both DC and cumulated DSR present different distributions for severe and weak years but differences between these two groups are more pronounced in the case of cumulated DSR (this is especially visible when taking into account the interquartile ranges, as indicated by the shaded areas in light red and light green).

Question: Finally, I think that much emphasis is placed on the "diagnostic model of BA", i.e. using meteorological information during the pre-fire and the fire seasons while this model does not bring much added value to the field.

Answer: The "diagnostic model of BA" (i.e. the model with pre-fire and fire season covariates, respectively psi(d) and chi) requires meteorological information along the fire season and its usefulness is therefore limited as a prediction tool of severity of the fire season. However, when pre-fire conditions are known (i.e. psi(d) at a given day), the diagnostic model may still be used to anticipate the severity of the fire season. This may be achieved by specifying a threshold of probability that a certain amount of BA will be exceeded, then inverting the model to compute the required value of chi (given the known value of psi(d)) and finally estimating the probability that this value will be exceeded in the fire season (e.g. based on the statistical distribution of chi of past years). This empirical line of reasoning was adopted in a previous feasibility study (Nunes et al., 2014). Here, the "diagnostic model of BA" is used as a benchmark to assess the decrease in performance of the statistical model of BA when reducing to the meteorological covariate respecting to the fire season. As shown in Fig.4, the relative importance of covariate psi(d) increases along the fire season and therefore the loss in performance decreases when reducing from covariates psi(d) and chi to covariate psi(d). Nunes, S. A., DaCamara, C. C., Turkman, K. F., Ermida, S. L. and Calado, T. J.: Anticipating the severity of the fire season in Northern Portugal using statistical models based on meteorological indices of fire danger, in: Advances in Forest Fire Research 2018 (Ed Domingos Xavier Viegas), edited by: Imprensa da Universidade de

Coimbra, ISBN 978-989-26-0884-6, 1634-1645, http://dx.doi.org/10.14195/978-989-26-0884-6_180, 2014.

Question: P2, L25-28: That part is interesting and should be developed. You should also provide some citations that shows the links between spring drought and the likelihood of summer heatwaves. Not sure the references are appropriate

Answer: The reviewer is correct when pointing out that the link between hot and dry spring with summer heatwaves was not clearly supported by appropriate references. The text was amended in order to make these connections more clear as follows: The rationale is that soil moisture deficit and drought have an impact on the increased frequency and can amplify the magnitude of hot summer extreme events in the Mediterranean (Vautard et al., 2007; Hirschi et al., 2011). Thus, persistent warm and dry conditions along the pre-fire season induce thermal and water stress on vegetation making the landscape more prone to the occurrence of very severe fire episodes and, at the same time, increase the likelihood of heat wave spells that steer the onset and propagation of large fires (Gudmundsson et al., 2014; Turco et al., 2017).

Question: P3, L15: could you provide more details on how does the DSR differs from FWI.

Answer: DSR results from a direct transformation of the Fire Weather Index (FWI), the last of the six components of CFFWIS, according to the relation DSR=0.0272 (FWI)^1.77. This transformation weights FWI sharply as it raises so that DSR becomes more suitable than FWI to be cumulated or averaged. These two sentences are now included in the manuscript.

Question: P4 : L8-14: That would be clearer if that part was moved before indices descriptions

Answer: The sentence was moved as suggested.

Question: P4 : L16, is psi(d) computed according to the mean and standard deviation

of day d or over the entire population (including all days)

Answer: psi(d) is obtained by normalizing D_pfs(d), i.e. by subtracting the mean and dividing the standard deviation of that day. The original text was slightly enlarged for clarification.

Question: P7, L7, why starting from May 26 only. That would be interesting to start earlier to see also when does that information becomes relevant

Answer: Indeed, the starting date of May 26th was set a posteriori so that results presented provide relevant information for users. This is now clarified in the manuscript.

Please also note the supplement to this comment:
https://www.nat-hazards-earth-syst-sci-discuss.net/nhess-2019-60/nhess-2019-60-AC2-supplement.pdf

––––––––––––––––––––––––––––

[Figure]

**Fig. 1.** Daily values of DC for the pre fire season for severe, moderate and weak years (red, black and green lines). The areas between percentiles 25 and 75 for the severe and weak years are also indicated.

**Fig. 2.** Daily values of DSR for the pre fire season for severe, moderate and weak years (red, black and green lines). The areas between percentiles 25 and 75 for the severe and weak years are also indicated.